# Airborne Exposure of the Cornea to PM_10_ Induces Oxidative Stress and Disrupts Nrf2 Mediated Anti-Oxidant Defenses

**DOI:** 10.3390/ijms24043911

**Published:** 2023-02-15

**Authors:** Mallika Somayajulu, Sharon A. McClellan, Robert Wright, Ahalya Pitchaikannu, Bridget Croniger, Kezhong Zhang, Linda D. Hazlett

**Affiliations:** 1Department of Ophthalmology, Visual and Anatomical Sciences, School of Medicine, Wayne State University, 540 E. Canfield, Detroit, MI 48201, USA; 2Center for Molecular Medicine and Genetics, Wayne State University School of Medicine, 540 E. Canfield, Detroit, MI 48201, USA

**Keywords:** particulate matter, Nrf2, SKQ1, cornea, mouse

## Abstract

The purpose of this study is to test the effects of whole-body animal exposure to airborne particulate matter (PM) with an aerodynamic diameter of <10 μm (PM_10_) in the mouse cornea and in vitro. C57BL/6 mice were exposed to control or 500 µg/m^3^ PM_10_ for 2 weeks. In vivo, reduced glutathione (GSH) and malondialdehyde (MDA) were analyzed. RT-PCR and ELISA evaluated levels of nuclear factor erythroid 2-related factor 2 (Nrf2) signaling and inflammatory markers. SKQ1, a novel mitochondrial antioxidant, was applied topically and GSH, MDA and Nrf2 levels were tested. In vitro, cells were treated with PM_10_ ± SKQ1 and cell viability, MDA, mitochondrial ROS, ATP and Nrf2 protein were tested. In vivo, PM_10_ vs. control exposure significantly reduced GSH, corneal thickness and increased MDA levels. PM_10_-exposed corneas showed significantly higher mRNA levels for downstream targets, pro-inflammatory molecules and reduced Nrf2 protein. In PM_10_-exposed corneas, SKQ1 restored GSH and Nrf2 levels and lowered MDA. In vitro, PM_10_ reduced cell viability, Nrf2 protein, and ATP, and increased MDA, and mitochondrial ROS; while SKQ1 reversed these effects. Whole-body PM_10_ exposure triggers oxidative stress, disrupting the Nrf2 pathway. SKQ1 reverses these deleterious effects in vivo and in vitro, suggesting applicability to humans.

## 1. Introduction

Air pollution is a major contributor to health problems worldwide [1]. The World Health Organization (WHO) air quality model demonstrates that ambient air pollution annually causes 4.2 million deaths, and 91% of the world’s population lives in places where air quality exceeds the limits of WHO guidelines [2]. Epidemiological evidence shows adverse health effects associated with exposure to airborne particulate matter with a mean aerodynamic diameter of <10 μm (PM_10_) [3]. Long-term exposure is associated with increased risk of cardiovascular [4], cerebrovascular [5], pulmonary diseases [6], arteriosclerosis [7], and cancer [8], while short-term exposure can lead to asthma [9], bronchitis [10], and other respiratory ailments [11]. The effects of PM_10_ have largely focused on the lungs [12,13,14]. However, very little is known about the effects of PM_10_ on the cornea, which is also readily exposed to this pollutant. Epidemiological and clinical data suggest that air pollution in which PM is a major constituent can cause transient ocular allergies (redness, discharge, foreign body sensation, and itching) [15]. Clinically, dry eye caused by particulates, is a global problem [16]. In this regard, studies from South Korea showed that high PM_10_ exposure correlated with increased outpatient visits for ocular diseases, including emergency room visits for keratitis [17]. A recent study in mice examined the effects of air pollution in Argentina and found that exposure to polluted air compromised corneal immunity and worsened inflammation in acute herpes simplex keratitis [18]. In fact, dry eye [19,20,21,22] and conjunctivitis [23,24] are linked (causative or makes the disease worse), in general, to the effects of air pollution increasing susceptibility to infection. Additionally, a new study in mice examined the effects of particulate matter in China and showed that exposure to particulates contributed to the initiation and advancement of ocular hypertension and glaucoma [25].

Exposure to PM_10_ induces oxidative stress [26] and the generation of free radical species [27]. This oxidative stress has been attributed to PM_10_ components including polycyclic aromatic hydrocarbons [28], ambient ultrafine particles [29] and transition metals such as iron [30]. In fact, metals present in particulate matter can enter mitochondria and perturb mitochondrial membrane potential and induce mitochondrial reactive oxygen species (ROS) and induce apoptosis [31]. Oxidative stress can activate nuclear factor erythroid-2-related factor 2 (Nrf2), a redox-sensitive transcription factor that regulates the expression of downstream anti-oxidant genes and phase-II enzymes that counter ROS and thus protect from adverse biological outcomes [32]. However, repeated and prolonged exposure to polycyclic aromatic hydrocarbons has been shown to inactivate the Nrf2 signaling pathway causing impaired mitochondrial redox homeostasis and leading to mitochondrial dysfunction [33]. In this regard, SKQ1 (10-(6′-plastoquinonyl) decyltriphenylphosphonium) is a mitochondria-specific anti-oxidant that can cross the plasma membrane and accumulate in the inner mitochondrial membrane where it is reduced or recharged in a controlled fashion and surpasses the efficacy of traditional anti-oxidants [34]. The conversion between the oxidized and reduced forms of SKQ1 helps mitigate the damage induced by mitochondrial ROS [35,36]. Studies have shown SKQ1 to be protective against damage from oxidative stress in animal models of ischemia/reperfusion [37], aging [38] and neurodegenerative diseases [39]. Additionally, SKQ1 has been formulated as an eye drop (Visomitin) in Russia and has been shown to prevent anesthesia-induced dry eye syndrome in patients who underwent long-term general anesthesia or ocular surgery [40]. Recently, it has been reported to have undergone a phase 3 clinical trial for the treatment of dry eye disease in the USA [41]. The aim of this study is to understand the effects of PM_10_ in vivo on the normal mouse cornea and in vitro in human corneal epithelial cells with an emphasis on the protective role of SKQ1 as an anti-oxidant and a cytoprotective agent with clinical potential.

## 2. Results

### 2.1. PM_10_ Exposure: Tear Secretion and Corneal Sensitivity

The effects of whole-body exposure to PM_10_ vs. control on tear secretion and corneal sensitivity at 0 (pre-exposure) and 2 weeks (post-exposure) are shown in Figure 1A–C. No differences in tear volume of PM_10_ vs. control exposed mice were detected at the 2 weeks time period as observed by the phenol-red thread images (Figure 1A) and tear volume represented as a bar graph in Figure 1B. No significant differences in corneal sensitivity were observed between PM_10_ vs. control exposed groups (Figure 1C) at 2 weeks of exposure.

### 2.2. Effects of PM_10_ on GSH and MDA Levels and Histopathology

Levels of GSH were significantly lower (*p* < 0.05) after 2 weeks of exposure to PM_10_ vs. control (Figure 2A). Levels of MDA (Figure 2B) were significantly increased (*p* < 0.05) in PM_10_ vs. control exposed corneas at 2 weeks. Because we saw biochemical changes in GSH and MDA levels at 2 weeks of exposure, we next examined paraffin-embedded and hematoxylin and eosin-stained corneas at that time period and the data are shown in Figure 2C. PM_10_ vs. control exposure showed that the epithelium of the cornea was compacted, nuclei in the superficial corneal layers were pynknotic, and infiltrating cells (containing a brownish particulate, Figure 2C inset) were observed in the PM_10_ exposed mice. These cells were easily distinguished due to their distinct cell borders which were visible between corneal epithelial cells which normally are joined by intercellular junctions, including tight junctions. PM_10_ vs. control significantly reduced the thickness of the epithelium (Figure 2D), stroma (Figure 2E) and the entire cornea (Figure 2F, *p* < 0.001 for all).

### 2.3. Effects of PM_10_ on Innate Immunity

mRNA levels of innate immune markers measured after 2 weeks of exposure to control vs. PM_10_ are shown in Figure 3A–G. mRNA levels were significantly elevated for chemokine (C-X-C) ligand (CXCL)2 (A, *p* < 0.001), Toll-like receptor (TLR)4 (B, *p* < 0.001), TLR2 (C, *p* < 0.05), interleukin (IL)-6 (D, *p* < 0.01), and IL-1β (E, *p* < 0.01) in PM_10_ vs. control exposed animals. No significant differences were observed in the levels of tumor necrosis factor (TNF)-α (F) and receptor for advanced glycation end products (RAGE) (G) in PM_10_ vs. control animals.

### 2.4. Response to PM_10_: Nrf2, GSH Maintenance enzymes and Effects of SKQ1 on Oxidative Stress

mRNA levels of GSH maintenance enzymes are shown in Figure 4A–D. Significantly elevated mRNA levels were observed for GSH maintenance enzymes: glutathione peroxidase (GPX) 1 (B, *p* < 0.01), GPX2 (C, *p* < 0.01) and glutathione reductase (GR)1 (D, *p* < 0.001); while Nrf2 (A) levels did not change in PM_10_ vs. control exposure at 2 weeks. Figure 4E–G indicates levels of GSH, MDA and Nrf2 protein after 2 weeks of PM_10_ vs. control exposure and the effects of the mitochondrial anti-oxidant SKQ1. Figure 4E shows a significant decrease (*p* < 0.01) in GSH levels after 2 weeks of PM_10_ vs. control exposure which are significantly reversed (*p* < 0.01) by SKQ1 treatment. MDA levels were significantly increased (*p* < 0.01) after PM_10_ vs. control exposure, and treatment with SKQ1 significantly decreased (*p* < 0.001) MDA levels. Nrf2 protein levels were significantly (<0.01) after PM1- vs. control exposure and SKQ1 restores Nrf2 levels almost to the control level (*p* < 0.05).

### 2.5. Effects of SKQ1 on Cell Viability, Oxidative Stress, Mitochondrial ROS, ATP and Nrf2 Levels after PM_10_ Exposure in HCET

Phase contrast images of HCET exposed to 100 μg/mL of PM_10_ in the presence or absence of SKQ1 are shown (Figure 5A–C). Figure 5A shows that in the media control group, without PM_10_, cells are spindle-shaped with prominent nuclei. PM_10_-exposed cells appear to be thinned with a few rounded cells, and the nuclei are difficult to see (Figure 5B). In the presence of SKQ1, PM_10_-exposed cells appear similar to the media control group, with spindle-shaped cells and prominent nuclei (Figure 5C). HCET exposed to 100µg/mL PM_10_ in the absence of SKQ1 showed significantly lower (*p* < 0.001) cell viability vs. media control (Figure 5D). Pre-treatment with SKQ1 significantly increased (*p* < 0.001) the viability of cells exposed to 100 µg/mL PM_10_. Viability was further reduced (*p* < 0.001) when cells were treated with 200 µg/mL PM_10_ vs. media control. However, at 200 µg/mL PM_10_, SKQ1 treatment was unable to positively affect cell viability. MDA levels were significantly increased (*p* < 0.01) in PM_10_ exposed vs. media control (Figure 5E). SKQ1 significantly reduced (*p* < 0.05) MDA levels in the PM_10_-exposed cells (Figure 5E). Figure 5F shows that the levels of mitochondrial ROS were significantly increased (*p* < 0.001) by PM_10_ vs. media control. Pre-treatment with SKQ1 significantly reduced (*p* < 0.001) levels of mitochondrial ROS (Figure 5F). PM_10_ vs. media control levels significantly lowered (*p* < 0.001) ATP (Figure 5G). SKQ1 significantly restored (*p* < 0.001) ATP levels in PM_10_-exposed cells (Figure 5G). The protein levels of Nrf2 measured by Western blot and the integrated density values (IDV) after normalizing Nrf2 to β-tubulin are represented in Figure 5H. Data indicate a significant reduction (*p* < 0.001) in PM_10_ exposed cells vs. media control. Pre-treating HCET with SKQ1 significantly restored (*p* < 0.001) Nrf2 levels after PM_10_ exposure (Figure 5H).

### 2.6. Effects of SKQ1 on Cell Viability and Nrf2 Levels after PM_10_ Exposure in HCEC

Since we used immortalized corneal epithelial cells (HCET) to study the effects of PM_10_, we next tested the toxic effects of PM_10_ on primary human corneal epithelial cells (HCEC). A significant reduction (*p* < 0.001) in HCEC cell viability was observed upon exposure to 100 µg/mL PM_10_ vs. media control (Figure 6A). SKQ1 pre-treatment significantly improved (*p* < 0.001) the viability after PM_10_ exposure in HCEC (Figure 6A). Western blot analysis showed that PM_10_ vs. media control significantly reduced (*p* < 0.001) the protein levels of Nrf2 (Figure 6B). SKQ1 pre-treatment significantly increased (*p* < 0.01) Nrf2 levels after PM_10_ exposure vs. PM_10_ without SKQ1 (Figure 6B).

## 3. Discussion

Exposure of the ocular surface to air pollutants can cause significant irritation and discomfort to the eye [24]. While eye diseases do not affect life expectancy, they nonetheless result in a significant reduction in the quality of life [24]. One such condition is dry eye disease, which is triggered by an array of factors, including tear film instability, and is often associated with elevated tear osmotic pressure and ocular inflammation [42]. Epidemiological studies have implicated the role of air pollution in dry eye disease [43] but the exact mechanism by which PM exerts its toxic effects on the cornea is still not completely understood. Different animal models have been developed to study the effects of PM on the eye using topical eye drops [44,45]; however, the topical application does not reflect actual environmental exposure to particulates. To overcome this limitation, we used a whole-body aerosol exposure system to disperse PM_10_, a well-characterized product purchased from the National Institute of Standards and Technology (NIST, SRM 2787) [46].

A study using PM_10_ drops at a very high concentration (5 mg/mL, 4X/day) showed reduced tear volume, damaged tear film, impaired ocular surface and reduced goblet cell number in the conjunctiva in male BALB/c mice [47]. However, when we examined the effects of PM_10_ exposure (500 μg/m^3^ in a whole body exposure chamber) on tear secretion, no significant change in tear volume was observed after 2 weeks of exposure. Our results also differ from a recent study in female rats exposed to PM in an exposure chamber (500 μg/m^3^ for 2 weeks) that showed a significant reduction in tear volume after 14 days [48]. These differences could be due to the type of particulates used (we used SRM 2787 vs. samples from China), the time of exposure (3 h/d vs. 5 h/d) and different species (mice vs. rats). Reduced corneal sensitivity is another characteristic of dry eye disease in patients [49]. However, we did not see any alterations in corneal sensitivity after 2 weeks of PM_10_ exposure. Very little is known about the effects of PM_10_ on corneal sensitivity in animal models. In our study, histological evaluation of PM_10_-exposed corneas showed pynknotic nuclei and infiltration of inflammatory cells in the epithelium of the cornea. Our data are similar to a previous study that showed increased apoptosis and the infiltration of inflammatory cells in the central cornea of mice that received drops of PM_10_ (5 mg/mL) for 2 weeks [47]. Furthermore, we observed that PM_10_-exposed corneas exhibited a reduction in the thickness of the epithelium, stroma and the whole cornea. These data are similar to a previous study that examined the effects of the long-term exposure of PM eye drops (5 mg/mL 4x/d) on the cornea, conjunctiva and retina in rats and found a decreased thickness of the epithelium and whole cornea [40]. To understand how PM_10_ exerts its toxicity on the cornea, we then evaluated levels of oxidative stress and inflammation. Our data showed increased pro-inflammatory cytokines and innate immune response markers in the mouse cornea after PM_10_ exposure. Similar effects of PM_10_ on pro-inflammatory cytokine levels have been reported in the eyes of mice that received PM_10_ eye drops for 2 weeks [47]. In addition, we also observed that PM_10_ induced oxidative stress (increased MDA levels) by disrupting the anti-oxidant capacity (decreased GSH levels) in the cornea. In this regard, increased lipid peroxidation and reduced GSH levels have been previously reported in the lungs of rats after PM_10_ instillation [50]. GSH maintenance and synthesis are regulated by the Nrf2 signaling pathway [51]. Nrf2 regulates the rate of GSH synthesis by controlling the activity of the enzymes required for synthesis namely γ-glutamyl cysteine ligase and glutathione synthetase [51] and also controls GSH maintenance by regulating GPX [52], and GR1 [53]. In our study, PM_10_ exposure increased the transcript levels of GPX1, GPX2 and GR1, while Nrf2 levels were unchanged in the mouse cornea after 2 weeks. In contrast, reduced protein levels of Nrf2 were observed which are consistent with data from a mouse model of experimental autoimmune encephalomyelitis (EAE) [54]. Interestingly, low Nrf2 protein levels in EAE were not due to a reduced amount of Nrf2 mRNA, which instead slightly increased in the diseased tissue with time. The discrepancy between mRNA and protein levels for Nrf2 was noted and could be due to alterations in translation and/or other unknown post-translational processes of Nrf2 under pathological conditions such as EAE [54].

Ongoing research is focused on developing eye therapeutics to combat the deleterious effects of air pollution on the eye [55,56]. SKQ1 (Visomitin), a novel mitochondria-targeted anti-oxidant has been used to treat inflammation linked to anesthesia-induced dry eye disease and corneal wounds [40]. In experimental models, SKQ1 administered for 5 days at 50 nmol/kg increased mRNA levels of Nrf2 and antioxidant enzyme genes (SOD1, SOD2, CAT, GPX4) in the cerebral cortex of rat brain under normal and hyperoxic conditions [57]. In our study, we tested the efficacy of SKQ1 as a protective agent against the deleterious effects induced by PM_10_ in the mouse cornea. SKQ1 protected the cornea against PM_10_ toxicity by significantly reducing oxidative stress (reducing lipid peroxidation) and restoring levels of anti-oxidant GSH and Nrf2.

Since data generated in a mouse system often does not translate to human applicability, we further tested the effects of PM_10_ and SKQ1 in vitro on transformed and primary human corneal epithelial cells. PM_10_ reduced cell viability in a concentration-dependent manner in transformed corneal epithelial cells. In this respect, PM has been shown to affect cell viability in vitro in a dose-dependent manner in human corneal epithelial cells [58]. We observed that anti-oxidant SKQ1 significantly reversed PM_10_-induced loss in cell viability. Our study showed that SKQ1 was protective when cells were treated with 100 μg/mL, but not 200 μg/mL of PM_10_. Therefore, we selected 100 μg/mL PM_10_ as the optimal dose for all further experiments. We further confirmed our findings in primary human corneal epithelial cells and observed that SKQ1 could reverse the toxic effects of PM_10_ on cell viability. The protective effects of SKQ1 on the ocular surface have been previously established [59]. SKQ1 can enhance cell proliferation and boost corneal wound healing in corneal limbal epithelial cells at a concentration of 50 nM [59]. Furthermore, the study indicated that at higher concentrations, SKQ1 was cytotoxic [59].

Oxidative stress and inflammation have been implicated as the main mechanisms of PM-mediated toxicity [58]. Components of PM such as poly aromatic hydrocarbons and heavy metals can generate reactive oxygen species (ROS) leading to oxidative stress. Mitochondria are the major generators of ROS, but they are also targets of PM toxicity [60]. In this regard, PM_2.5_ can accumulate in the mitochondria, disrupt mitochondrial structure and function, induce mitochondrial ROS and activate the intrinsic apoptotic pathway [32,61]. When we tested PM_10_ toxicity in transformed human corneal epithelial cells; we observed increased lipid peroxidation (MDA), increased mitochondrial ROS and reduced ATP production after PM_10_ treatment. Impaired mitochondrial function, amplified ROS and increased apoptosis have been previously reported in lung epithelial cells treated with oil fly ash [61] or PM samples collected in Milan [62]. Oxidative stress and inflammation induced by acute or chronic exposure to particulates involve Nrf2 signaling [32,63,64]. At low doses or single acute exposure, PM induces Nrf2 expression [63] but at high doses or multiple chronic exposures, may cause an oxidative burst and thus compromise Nrf2 signaling [32,64]. This can acutely damage mitochondrial redox balance and reduce energy levels [32,65]. We treated both transformed and primary human corneal epithelial cells with a high dose (100µg/mL) of PM_10_ and observed a decrease in Nrf2 protein levels. Our data are similar to a study in human alveolar epithelial cells which showed a significant reduction in Nrf2 levels and impairment of GSH after treating cells with 100–500 μg/mL of PM_10_ [50]. Additionally, SKQ1 protected cells from oxidative stress, and mitochondrial dysfunction and restored Nrf2 levels after PM_10_ exposure. In conclusion, we have shown (diagrammatically in Figure 7) that whole-body exposure to PM_10_ triggers oxidative stress and disrupts the Nrf2 pathway in the cornea. SKQ1 treatment reverses these deleterious effects. In vitro human corneal epithelial cell data parallel the in vivo effects.

## 4. Materials and Methods

### 4.1. Mice

Eight-week-old C57BL/6 female mice were purchased from the Jackson Laboratory (Bar Harbor, ME, USA) and housed in accordance with the National Institutes of Health guidelines. They were humanely treated and in compliance with both the ARVO Statement for the Use of Animals in Ophthalmic and Vision Research) and the Institutional Animal Care and Use Committee of Wayne State University (IACUC 21-09-4042).

### 4.2. Whole Body Exposure to PM_10_

Particulate matter, a major air pollutant is a general term for small atmospheric solid and liquid particles that vary in size (2.5–10 μm), composition and origin [66]. All experiments in this study were performed with PM_10_ purchased from the National Institute of Standards and Technology, (NIST) (Standard Reference Material (SRM) 2787). A whole-body exposure chamber (CH Technologies, Westwood, NJ, USA) was used for all the in vivo experiments performed in this study (Figure 8). Exposures were carried out in a stainless steel, whole-body inhalation chamber; one group was exposed to PM_10_ whereas the other received only humidified compressed air (control). The apparatus is equipped with a Vilnius aerosol generator (VAG), a dry powder dispenser that offers high stability of aerosol output. Mice were exposed to control or an acute, high dose of PM_10_ (500 µg/m^3^) for 2 weeks for 3 h/day/5 days/week and rested on the weekends. For the in vivo study, the dosage was based on mean PM_10_ concentrations measured in the winter, ranging as high as 494 µg/m^3^ in five Chinese cities [67]. For in vitro studies, PM_10_ was used at a concentration of 100 μg/mL as previously published [68,69].

### 4.3. SKQ1 Treatment

SKQ1 (BOC Sciences, Shirley, NY, USA) was used to treat PM_10_-exposed mice using a published dose of 7.5 µM [40]. Eyes were treated topically with 5 μL SKQ1 or PBS (control), three times a day before the first chamber exposure and then once each day before exposure for 2 weeks. For in vitro studies, SKQ1 was used at a concentration of 50 nM as previously described [59].

### 4.4. Endotoxin and Beta-D-Glucan

We are using Standard Reference Material (SRM) 2787 National Institute of Standards and Technology, (NIST) which is one of the most characterized particulate materials for a wide range of organic and inorganic constituents [46].

The mass fractions of the constituents represent those in a contemporary urban environment and complement the mass fractions found in existing SRM. To determine levels of endotoxin, we also tested SRM 2787 using the Pyrochrome Kit for detection and quantification of Gram-negative bacterial endotoxin (Associates of Cape Cod Inc. Falmouth, MA, USA) and endotoxin (*E. coli* O113:H10) to create a standard curve (ACCI). PM_10_ was tested at concentrations of 5 μg/mL, 1 μg/mL, 0.5 μg/mL (equivalent to 500 μg/m^3^, used for whole-body exposure), 0.25 μg/mL, and 0.125 μg/mL. All concentrations tested were below the limit of detection (0.005 EU/mL). We also tested for fungal contaminant (1,3)-Beta-D-Glucan using the Glucatell (1,3)-Beta-D-Glucan Kit (Associates of Cape Cod Inc.). PM_10_ was tested at the same concentrations used for testing endotoxin. All concentrations were below the limit of detection (0.625 pg/mL β-glucan standard, not shown). These data assure PM_10_ from NIST is not contaminated by either bacterial or fungal components which would complicate data analysis.

### 4.5. Tear Secretion

Tear levels were measured in the right eye of the control and PM_10_ exposed mice (*n* = 5/group/treatment) using Zone-Quick phenol red treated threads (Tianjin Jingming New Technological Development Co., Ltd., Tianjin, China) as described before [70]. Measurements were made before chamber exposure (week 0) to acquire a baseline reading and then at 2 weeks of exposure. Briefly, mice were lightly anesthetized and held while tear levels were measured by placing the tip of a phenol red impregnated cotton thread with a blunt tip on the bulbar conjunctiva (lateral canthus for 30 s). Appropriate care was taken to avoid touch stimulation of eyelashes and whiskers. The wetted length (in millimeters), indicated by a color change from yellow to red, was photographed and measured. Data are shown as mean wetted length + SD.

### 4.6. Corneal Sensation

Corneal sensitivity was measured for control and PM_10_ exposed mice (*n* = 5/group/treatment) using a Cochet–Bonnet esthesiometer (Luneau SAS #10129; Prunay-le-Gillon, France) as described before [70]. Measurements were made before chamber exposure (week 0) to acquire a baseline reading and at 2 weeks after exposure. Measurements were made from a filament length of 60 mm and gradually decreased in 5 mm steps to a length of 5 mm. For each filament length, five repeat measurements were conducted. If three blinks were evoked in response to five consecutive touches, a positive response was recorded. The longest filament length which caused a positive response was recorded as the corneal sensitivity threshold. Data are shown as mean filament length + SD.

### 4.7. Reduced Glutathione (GSH) Assay

Total GSH levels were analyzed by a glutathione assay kit (Cayman Chemical, Ann Arbor, MI, USA) per the manufacturer’s protocol and as previously described [71]. Briefly, individual mouse corneas (*n* = 5/group/treatment) from control vs. PM_10_ exposed groups ± SKQ1 were harvested in 500 µL of ice-cold 50 mM MES (2-(N-morpholino) ethanesulphonic acid) containing 1 mM EDTA, homogenized and centrifuged at 10,000× *g* at 4 °C for 10 min and the supernatant was collected. Total GSH levels were determined using Ellman’s reagent (DTNB, 5,5′-dithio-bis-2-nitrobenzoic acid), which reacts with the sulfhydryl group of GSH resulting in a yellow-colored 5-thio-2-nitrobenzoic acid (TNB) product. Measurement of the absorbance of TNB at 405 nm provides an estimate of GSH in the sample. GSH levels were calculated using a standard curve and then normalized to the total protein in each sample. A Bradford assay was used to determine protein concentration in each sample. Protein levels were calculated using a standard curve with bovine serum albumin and Bio Rad protein assay reagent (Bio Rad, Hercules, CA, USA) per the manufacturer’s protocol. Final GSH levels were expressed as mean + SD.

### 4.8. ELISA

ELISA kits were used to measure protein levels of Nrf2 (Novus Biologicals, Centennial, CO, USA) and lipid peroxidation end product MDA (Cell Biolabs, San Diego, CA, USA). Briefly, individual mouse corneas (*n* = 5/group/treatment) from the control vs. PM_10_ exposed groups ± SKQ1 were harvested in 500 µL of PBS containing 0.1% Tween 20 and protease inhibitors. All assays were run per the manufacturer’s protocol and data were normalized to total protein (described above) and expressed as mean + SD.

### 4.9. H&E Staining of Exposed Corneas

Whole eyes (*n* = 3) were harvested from mice after 2 weeks of exposure to control or 500 μg/m^3^ PM_10_. Eyes were fixed in alcoholic z-fix and sent to Excalibur Pathology, Inc (Norman, OK) where they were embedded in paraffin, sectioned and stained with H&E. Photographs of representative areas of the epithelium were taken using a Leica DM4000B light microscope at 20X. The thickness of the whole cornea, epithelium and stroma were analyzed from three mice in each group (72 measurements/group/treatment). Data are represented as thickness (in mm) and expressed as mean + SD.

### 4.10. RT-PCR

Total RNA was isolated from control and PM_10_-exposed mouse corneas (*n* = 5/group/treatment) (RNA STAT-60; Tel-Test, Friendswood, TX, USA) per the manufacturer’s instructions as reported before [72]. Briefly, 1 μg of each RNA sample was reverse transcribed using Moloney-murine leukemia virus (M-MLV) reverse transcriptase (Invitrogen, Carlsbad, CA, USA) to produce a cDNA template and diluted 1:20 using DEPC-treated water. A 2 μL aliquot of diluted cDNA was used for the RT-PCR reaction. SYBR green/fluorescein PCR master mix (Bio-Rad Laboratories, Richmond, CA, USA) and primer concentrations of 10 μM were used in a total 10 μL volume. After a pre-programmed hot start cycle (3 min at 95 °C), PCR amplification was repeated for 45 cycles with parameters: 15 s at 95 °C and 60 s at 60 °C. Levels of chemokine CXCL2, TLR4, TLR2, IL-6, IL-1β, TNF-α, RAGE, Nrf2, GPX1, GPX2, and GR1 were tested by real-time RT-PCR (CFX Connect real-time PCR detection system; Bio-Rad Laboratories). The fold differences in gene expression were calculated relative to housekeeping gene β-actin mRNA and expressed as the relative mRNA concentration + SD. Primer pair sequences used are shown in Table 1.

### 4.11. Tissue Culture

HCET cells (HCE-2 [50.B1], ATCC, Gaithersburg, MA, USA) were cultured in Keratinocyte-serum free medium (Gibco, Grand Island, NY, USA) with 5 ng/mL human recombinant EGF, 0.05 mg/mL bovine pituitary extract, 0.005 mg/mL insulin, and 500 ng/mL hydrocortisone as reported before [73]. HCEC (primary corneal epithelial cells, ATCC) were cultured in Keratinocyte-serum free medium (Gibco) with 5 ng/mL human recombinant EGF and 0.05 mg/mL bovine pituitary extract as per the manufacturer’s protocol. To evaluate the effects of SKQ1, a subset of cells was incubated with 50 nM SKQ1 as previously described [59], 1 h prior to PM_10_ exposure. Phase contrast microscopy was used to photograph cell preparations using a Leica EL 6000 microscope (Deerfield, IL, USA).

### 4.12. Cell Viability Assay

An MTT 3-(4, 5-dimethylthiazol-2-yl)-2,5-diphenyltetrazolium bromide (ThermoFisher Scientific, Grand Island, NY, USA) assay was used to test the effects of PM_10_ on HCET viability in the presence and absence of SKQ1, per the manufacturer’s protocol and as reported before [73]. Briefly, 15,000 HCET cells were seeded in a 96-well plate, treated with increasing concentrations of PM_10_ (0, 100 and 200 µg/mL) ± 50 nM SKQ1 and incubated for 24 h. At the end of the treatments, 5 mg/mL MTT reagent was added to each well and incubated at 37 °C for 4 h and the media was removed. Dimethyl sulfoxide (DMSO) was added (50 µL/well) and optical density was read at 540 nm using a SpectraMax M5 microplate reader (Molecular Devices, Sunnyvale, CA, USA). The cell viability was tested similarly for HCEC with 0 and 100 µg/mL PM_10_ ± 50 nM SKQ1 after 24 h incubation. Data are shown as % cell viability + SD.

### 4.13. Mitochondrial ROS Assay

Mitochondrial ROS was analyzed by fluorescence with MitoSOX assay (ThermoFisher Scientific) according to the manufacturer’s protocol and as previously described [63]. Briefly, 15,000 HCET cells were plated onto 96 well plates, and treated with 100 µg/mL PM_10_ ± 50 nM SKQ1 for 6 h. Cells were washed with PBS and incubated with 10 µM MitoSOX at 37 °C for 10 min. Cells were then washed twice with PBS and fluorescence was measured at 510 nm (excitation) and 590 nm (emission) using SpectraMax M5 spectrophotometer (Molecular Devices). The level of ROS is proportional to fluorescence intensity. Data were normalized to control and are represented as % relative fluorescence + SD.

### 4.14. Mitochondrial ToxGlo Assay

To determine whether PM_10_ caused a reduction in ATP levels, a mitochondrial ToxGlo assay (Promega, Madison, WI, USA) was performed per the manufacturer’s protocol and as previously described [74]. Briefly, 15,000 HCET were plated onto 96 well plates, incubated overnight and media removed. Cells were then incubated galactose supplemented RPMI 1640 media and treated with 100 µg/mL PM_10_ ± 50 nM SKQ1 for 3 h. ATP detection reagent was added to each well, mixed and incubated for 5 min and luminescence was measured. The levels of ATP are represented as relative luminescence. Data are shown as mean + SD.

### 4.15. Western Blot

HCET or HCEC were treated with 100 µg/mL PM_10_ ± SKQ1 for 24 h, washed with ice-cold 0.1 M PBS (pH 7.4), lysed in RIPA buffer with protease and phosphatase inhibitors (SantaCruz Biotech, Dallas, TX, USA), incubated on ice for 20 min, centrifuged at 12,000× *g* at 4 °C for 10 min and the supernatant was collected. Total protein was determined from the supernatants using a BCA protein kit (ThermoFisher Scientific). Briefly, samples (75 μg) were run on SDS-PAGE in Tris-glycine-SDS buffer and electro-blotted onto nitrocellulose membranes (BioRad). After blocking for 1 h in 5% MTBST (Tris Buffer Saline containing 0.05% Tween 20 (TBST) and 5% nonfat milk), membranes were probed with primary antibodies: rabbit anti-mouse Nrf2 (1:500; Cell Signaling Technology, Danvers, MA, USA) in 2% MTBST overnight at 4 °C. After three washes with TBST, membranes were incubated with HRP-conjugated anti-rabbit secondary antibody (1:2000; Cell Signaling Technology) and diluted in 5% MTBST at room temperature for 2 h. Bands were developed with Supersignal West Femto Chemiluminescent Substrate (ThermoFisher Scientific), visualized using an iBright™ CL1500 Imaging System (ThermoFisher Scientific), and normalized to β-Tubulin (1:1000; Abcam, Waltham, MA, USA) and the intensity was quantified using ImageJ software. Data are shown as mean integrated density values (IDV) + SD.

### 4.16. Statistical Analysis

For in vivo data analysis, a one-way ANOVA followed by Bonferroni’s multiple comparison test (GraphPad Prism) was used to test the significance of tear production, corneal sensitivity, GSH, and ELISA (Nrf2, MDA). A Student’s *t*-test (GraphPad Prism) was used to determine the significance of the thickness of the whole cornea, epithelium, stroma, RT-PCR, GSH assay and ELISA (MDA) in vivo. For analyzing in vitro data, a one-way ANOVA followed by Bonferroni’s multiple comparison test (GraphPad Prism) was used to test the significance for MTT, ROS, ATP assay, ELISA (MDA) and Western blot. Data were considered significant at *p* < 0.05. All experiments were repeated at least once to ensure reproducibility and data are shown as mean + SD.

## Figures and Tables

**Figure 1 ijms-24-03911-f001:**
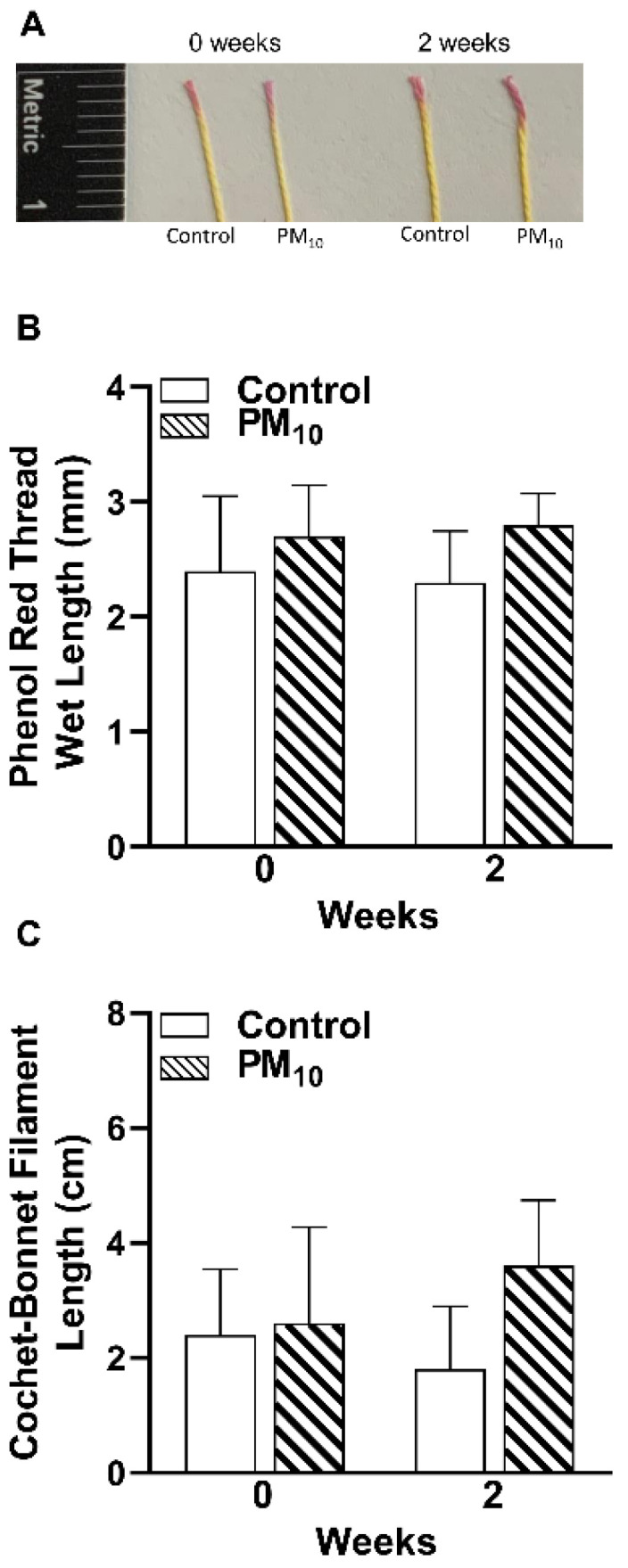
Effects of PM_10_ exposure on tear secretion and corneal sensitivity. (**A**) Phenol red thread images. Red portion of the thread represents tear volume at 0 and 2 weeks after control or PM_10_ exposure. (**B**) The wetted length (red portion) was measured and represented as a bar graph, showed no differences in tear volume. (**C**) Corneal sensitivity was measured using a Cochet-Bonnet Esthesiometer, showed no significant changes between control and PM_10_ groups. Data are expressed as mean + SD. (*n* = 5/group/time).

**Figure 2 ijms-24-03911-f002:**
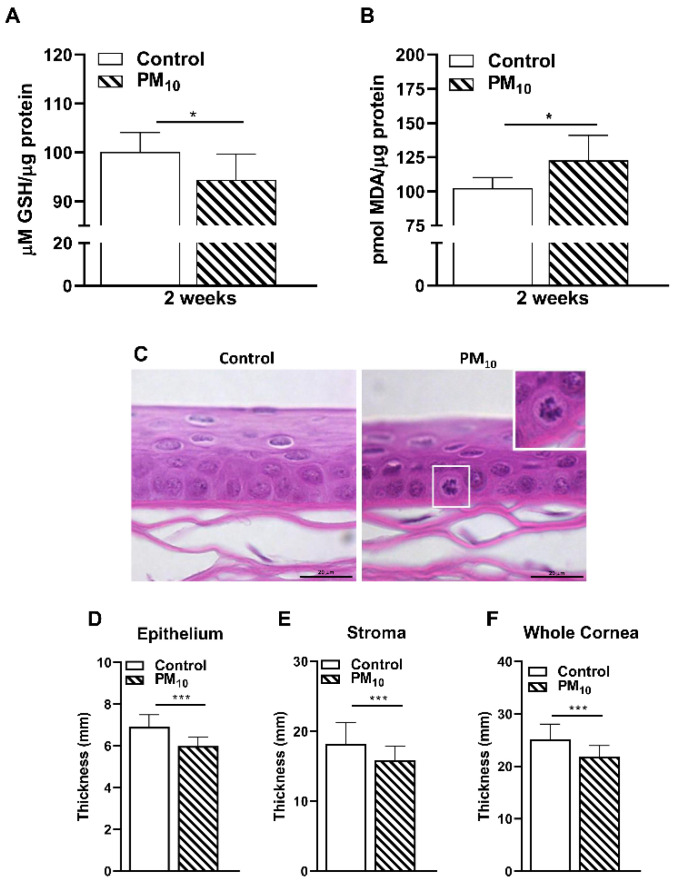
Effects of PM_10_ exposure: oxidative stress and histology. (**A**) Levels of GSH were significantly lower after 2 weeks exposure to PM_10_ vs. control. (**B**) MDA levels (lipid peroxidation end product) were significantly increased in PM_10_ vs. control exposed corneas after 2 weeks. (**C**) Paraffin embedded and hematoxylin and eosin stained corneas after 2 weeks exposure to PM_10_ or control. Inset shows an infiltrating cell containing a brownish particulate. Scale bar = 20 μm. Thickness of the epithelium (**D**), stroma (**E**) and the whole cornea (**F**) were significantly decreased in PM_10_ vs. control exposed corneas. Data are expressed as mean + SD. (* *p* < 0.05, *n* = 5/group (**A**,**B**) *** *p* < 0.001, *n* = 3/group (**D**–**F**)).

**Figure 3 ijms-24-03911-f003:**
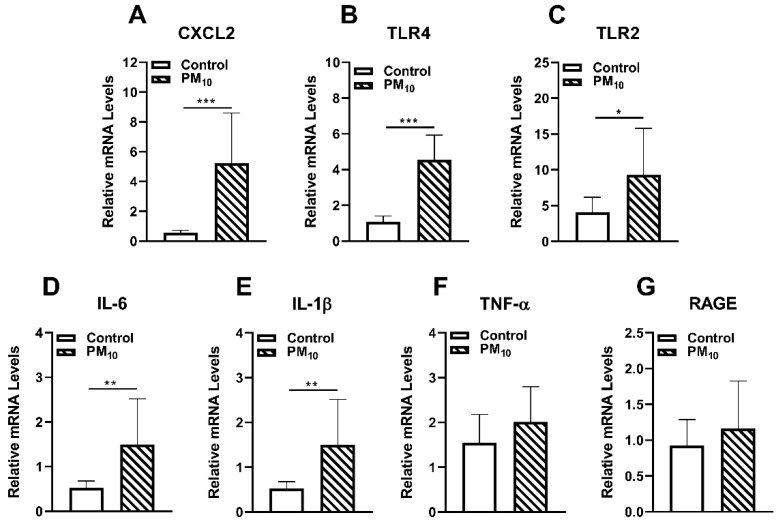
Effects of PM_10_ on innate immunity. RT-PCR showed significantly elevated mRNA levels in PM_10_ vs. control exposed corneas for chemokine CXCL2 (**A**), TLR4 (**B**), TLR2 (**C**), IL-6 (**D**), IL-1β (**E**) only but not TNF-α (**F**) and RAGE (**G**) after 2 weeks exposure. Data are expressed as mean + SD. (* *p* < 0.05, ** *p* < 0.01, *** *p* < 0.001, *n* = 5/group).

**Figure 4 ijms-24-03911-f004:**
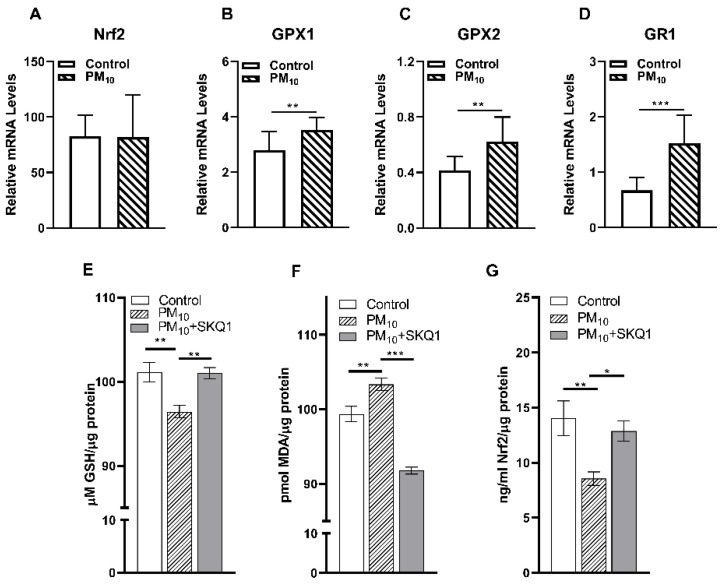
Effects of PM_10_ on Nrf2, GSH maintenance enzymes; protective effects of SKQ1. mRNA levels for GSH maintenance enzymes: GPX1 (**B**), GPX2 (**C**) and GR1 (**D**) are significantly increased in PM_10_ vs. control exposure at 2 weeks; no change for Nrf2 (**A**) levels. Effects of mitochondrial anti-oxidant SKQ1 on the levels of GSH, MDA and Nrf2 proteins after 2 weeks of PM_10_ vs. control exposure (**E**–**G**). SKQ1 protects corneas by significantly increasing GSH (E), reducing MDA (F) and restoring Nrf2 (G) levels. Data are expressed as mean + SD. (* *p* < 0.05, ** *p* < 0.01, *** *p* < 0.001, *n* = 5/group).

**Figure 5 ijms-24-03911-f005:**
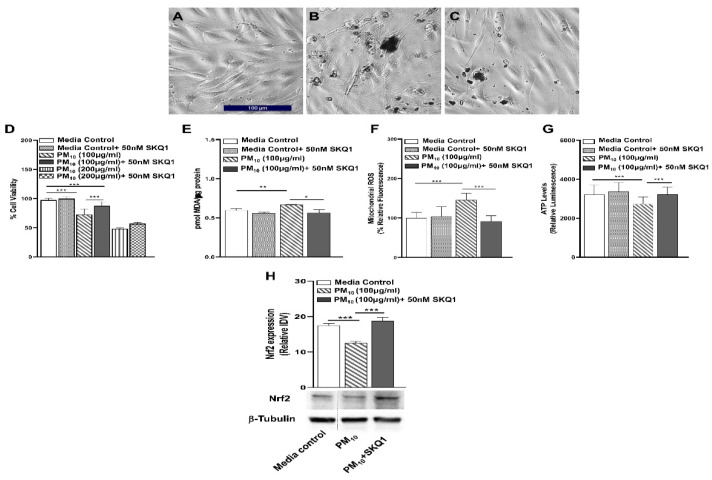
In vitro effects of SKQ1 on transformed HCET: morphology, viability, oxidative stress, Nrf2 protein, mitochondrial ROS and ATP after PM_10_ exposure. (**A**) Phase contrast images of cells in the media control group show elongated cells with prominent nuclei. (**B**) Cells exposed to PM_10_ are thinned and a few rounded cells are seen. (**C**) SKQ1 pre-treated cells appear spindle-shaped with prominent nuclei, similar to media control. Scale bar = 100 μm. (**D**) Exposure to 100 and 200 µg/mL PM_10_ significantly decreased cell viability vs. media control. SKQ1 pretreatment significantly increased cell viability only at 100 µg/mL but not at 200 µg/mL PM_10_. (**E**) MDA levels were significantly reduced by SKQ1 after exposure to PM_10_. (**F**) Elevated mitochondrial ROS after PM_10_ exposure was significantly reduced by SKQ1. (**G**) Reduced ATP levels due to PM_10_ exposure were restored by SKQ1. (**H**) Western blot analysis of Nrf2 protein showed that SKQ1 significantly restored Nrf2 levels after PM_10_ exposure. Data are expressed as mean + SD. (* *p* < 0.05, ** *p* < 0.01, *** *p* < 0.001, *n* = 3).

**Figure 6 ijms-24-03911-f006:**
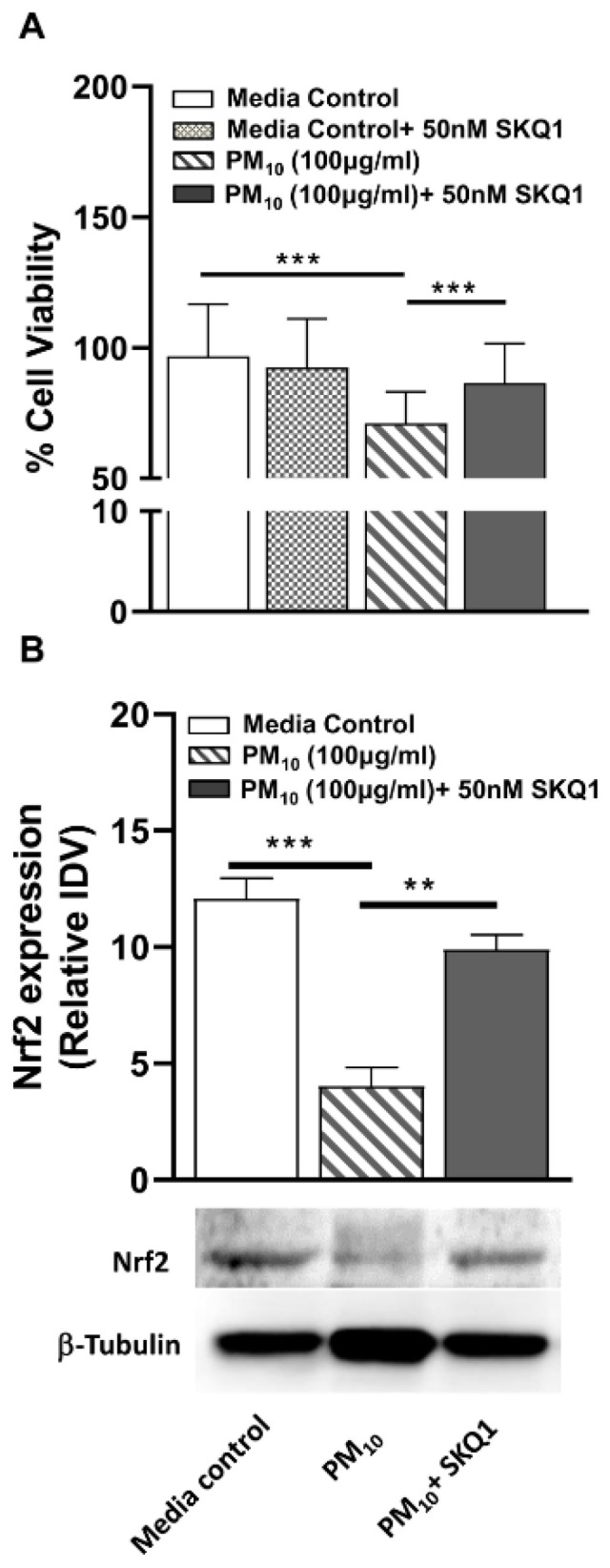
In vitro effects of SKQ1 on cell viability and Nrf2 levels after PM_10_ exposure in primary human corneal epithelial cells (HCEC). (**A**) Loss in cell viability due to PM_10_ exposure was significantly restored by SKQ1 pretreatment. (**B**) Western blot analysis for Nrf2 showed SKQ1 significantly increased Nrf2 levels after PM_10_ exposure. Data are expressed as mean + SD (** *p* < 0.01, *** *p* < 0.001, *n* = 3).

**Figure 7 ijms-24-03911-f007:**
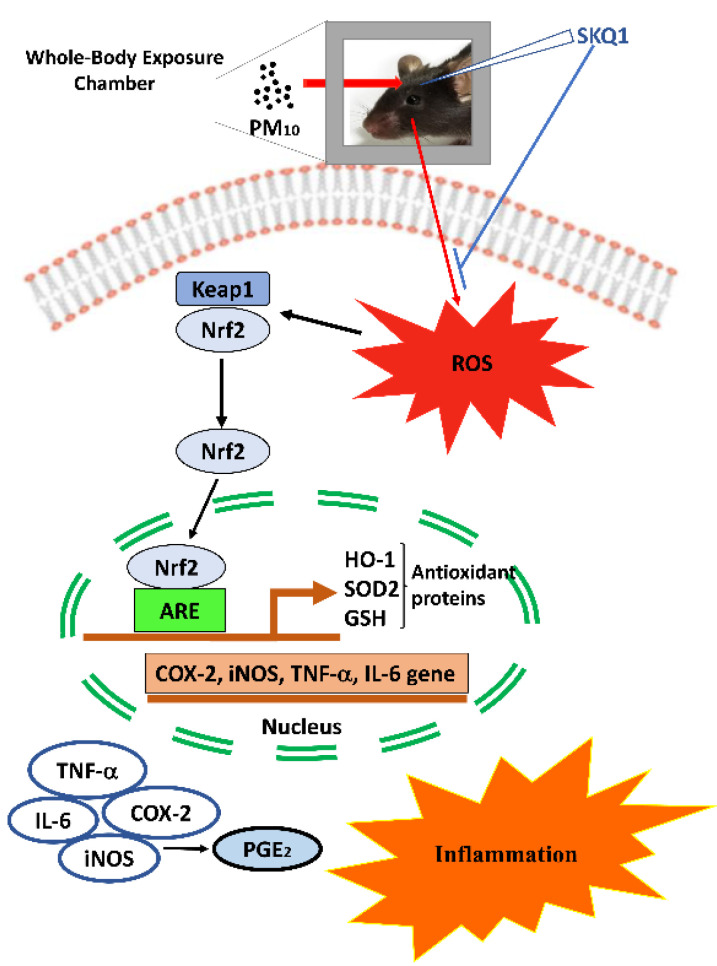
Whole body exposure to PM_10_ is shown diagrammatically.

**Figure 8 ijms-24-03911-f008:**
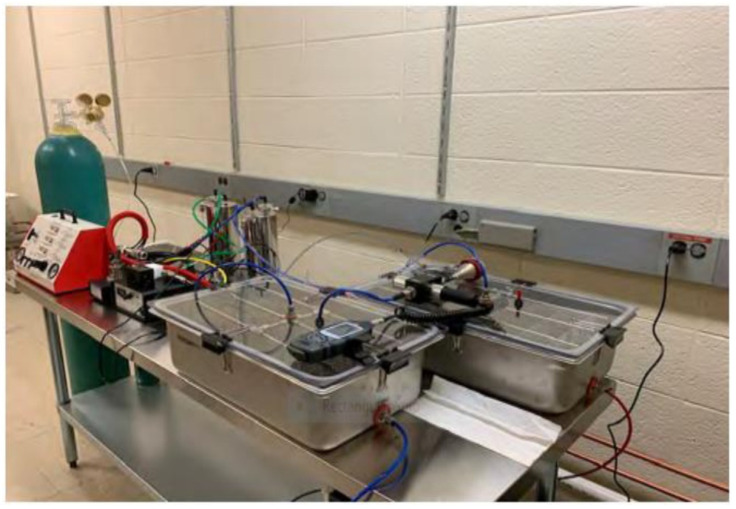
Airborne, whole body particulate matter exposure chamber.

**Table 1 ijms-24-03911-t001:** Nucleotide sequence of the specific primers used for PCR amplification (Mouse).

Gene	Nucleotide Sequence	Primer	GenBank
*Β-actin*	5′-GAT TAC TGC TCT GGC TCC TAG C-3′	F	NM_007393.3
	5′-GAC TCA TCG TAC TCC TGC TTG C-3′	R	
*Tlr2*	5′-CTC CTG AAG CTG TTG CGT TAC-3′	F	NM_011905.3
	5′-TAC TTT ACC CAG CTC GCT CAC TAC-3′	R	
*Tlr4*	5′-CCT GAC ACC AGG AAG CTT GAA-3′	F	NM_021297.2
	5′-TCT GAT CCA TGC ATT GGT AGG T-3′	R	
*Il-1β*	5′-TGT CCT CAT CCT GGA AGG TCC ACG-3′	F	NM_008361.3
	5′-TGT CCT CAT CCT GGA AGG TCC ACG-3′	R	
*Cxcl2*	5′-TGT CAA TGC CTG AAG ACC CTG CC-3′	F	NM_009140.2
(*Mip2*)	5′-AAC TTT TTG ACC GCC CTT GAG AGT GG-3′	R	
*Gpx1*	5′CTC ACC CGC TCT TTA CCTTCC T-3′	F	NM_008160.6
	5′-ACA CCG GAG ACC AAA TGA TGT ACT-3′	R	
*Gpx2*	5′-GTG GCG TCA CTC TGA GGA ACA-3′	F	NM_030667
	5′-CAG TTC TCC TGA TGT CCG AAC TG-3′	R	
*Gr1*	5′-CCA CGG CTA TGC AAC ATT CG-3′	F	NM_010344.4
	5′-GAT CTG GCT CTC GTG AGG AA-3′	R	
*Il-6*	5′-CAC AAG TCC GGA GAG GAG AC-3′	F	NM_031168.1
	5′-CAG AAT TGC CAT TGC ACA AC-3′	R	
*Nrf2*	5′-TGC CCC TCA TCA GGC CCA GT-3′	F	NM_010902.5
	5′-GCT CGG CTG GGA CTC GTG TT-3′	R	
*Rage*	5′-AGG CGT GAG GAG AGG AAG GCC-3′	F	NM_007425.2
	5′-TTA CGG TCC CCC GGC ACC AT-3′	R	
*Tnf-a*	5′-ACC CTC ACA CTC AGA TCA TCT T-3′	F	NM_013693.2
	5′-GGT TGT CTT TGA GAT CCA TGC-3′	R	

F, forward, R, reverse.

## Data Availability

Not applicable.

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
