# Peer review of "Airborne Exposure of the Cornea to PM10 Induces Oxidative Stress and Disrupts Nrf2 Mediated Anti-Oxidant Defenses"

_ijms, 2023, doi:10.3390/ijms24043911_

Round 1
Reviewer 1 Report
Well written and presented.
Author Response
Reviewer 1: Well written and presented.
Answer: We appreciate the positive response to our work and thank you.
Reviewer 2 Report
In this work, the authors studied the impact of PM10 exposure on cornea and reported the immune response and oxidative stress involved. The manuscript has made some unique points that have clinical implications. From the perspective of academic criticism, several technical concerns need to be addressed to further improve the quality of this manuscript, as appended below.
1. The reviewer would suggest to move Figure 7 to Figure 1 and add an schematic showing how the PM10-exposure animal model was built.
2. Figure 1A should be supplied with quantitative data, such as a table chart showing the measurement from multiple samples.
3. How was the dose/concentration of PM10 defined in the in vivo and in vitro experiments? Please add more details or references in the Results or the Discussion.
4. How was the dose/concentration and the duration of the SKQ1 treatment defined in the experiments?
5. For measuring the expression level, can the authors please explain why qPCR and western/ELISA was used separately in different figure? It would be more convincing if both the mRNA and protein expression were shown simultaneously.
6. It would be better for the presentation if the author can add some bright-field or phase contract images to show the change of corneal epithelial cell morphology with or without PM10 and SKQ1 treatment, in parallel to the data in Figure 4&5.
Author Response
Reviewer 2: Thank you for your helpful comments in review of our paper. The following is a point-by-point response to the review. The changes are highlighted in the manuscript.
1.The reviewer would suggest move Fig. 7 to Figure 1.
Answer: We appreciate your suggestion, however, because the Journal is constructed to show Methods after the main body of the manuscript, we will keep the figure where it is for appropriate flow of the manuscript. But, we have added a schematic in the discussion to show how the system works.
- Figure 1A should be supplied with quantitative data showing measurement from multiple samples.
Answer: Fig. 1A is currently followed by quantitative data (Fig. 1b n=5 mice). Fig 1c is similarly measured but tests sensitivity of the cornea. Thus, the measurement from several samples is shown. All data is repeated at least once and shown as the mean +Standard Deviation.
- How was the dose /concentration of PM10 defined in vivo and in vitro.
Answer: We have added more information to the Methods section and Discussion.
- How was the dose/concentration and duration of SKQ1 treatment defined in the experiments?
Answer: We have added information within the Methods and Discussion sections.
- For measuring the expression level, can the authors please explain why qPCR and western/Elisa were used separately in different figure. More convincing if shown simultaneously.
Answer: In Fig 3 we provide expression levels of several host innate immune factors but do not have protein. This was an indicator that besides oxidative stress, we have also activated the immune system. We show in Fig. 4 the effects on Nrf2 and GSH maintenance enzymes at the RNA level. And the protein shown in Fig. E, F, and G reflects those changes to Nrf2 protein and oxidative stress markers.
- It would be better for the presentation if the authors can add some bright-field or phase contrast images to show the change in corneal epithelial cell morphology with and w/o PM10 and SKQ1 in parallel to data in Fig 4 and 5.
Answer: Since we have H&E showing the in vivo cornea, and the changes in the epithelium, we have added a phase contrast image of HCET cells to reflect those changes to the cornea in the human cell line and the effects of PM10 +/-SKQ1 in vitro.
Reviewer 3 Report
This is a very interesting study, well conducted and with great interest for the future. Authors demonstrated how PM exposure could affect the cornea. The authors have tested a component (SKQ1) in mice with interesting outcomes. I really hope that in the near future they will be able to validate it in a human RCT. I really liked the work and I would like to congratulate the authors for their effort.
Minor changes:
Describe PM in the introduction
Author Response
Reviewer 3: We appreciate the positive response to our work and thank you.
Answer: We have added a few sentences to the Methods describing PM as requested. The changes are highlighted in the manuscript!
Round 2
Reviewer 2 Report
Accept in present form